# Epilepsy in a melanocyte-lineage mTOR hyperactivation mouse model: A novel epilepsy model

Fei Yang©, Lingli Yang©, Mari Wataya-Kaneda◉*, Lanting Teng, Ichiro Katayama

Department of Dermatology, Course of Integrated Medicine, Graduate School of Medicine, Osaka University, Osaka, Japan

© These authors contributed equally to this work.

* mkaneda@derma.med.osaka-u.ac.jp

**Data Availability Statement:** All relevant data are within the manuscript and its Supporting Information files.

## Abstract

### Objective

To clarify the complex mechanism underlying epileptogeneis, a novel animal model was generated.

### Methods

In our previous research, we have generated a melanocyte-lineage mTOR hyperactivation mouse model (*Mitf-M*-Cre *Tsc2* KO mice; cKO mice) to investigate mTOR pathway in melanogenesis regulation, markedly reduced skin pigmentation was observed. Very unexpectedly, spontaneous recurrent epilepsy was also developed in this mouse model.

### Results

Compared with control littermates, no change was found in either brain size or brain mass in cKO mice. Hematoxylin staining revealed no obvious aberrant histologic features in the whole brains of cKO mice. Histoimmunofluorescence staining and electron microscopy examination revealed markedly increased mTOR signaling and hyperproliferation of mitochondria in cKO mice, especially in the hippocampus. Furthermore, rapamycin treatment reversed these abnormalities.

### Conclusions

This study suggests that our melanocyte-lineage mTOR hyperactivation mouse is a novel animal model of epilepsy, which may promote the progress of both epilepsy and neurophysiology research.

**Funding:** This work was supported from a grant from the Ministry of Education, Culture, Sports, Science, and Technology of Japan (16k10155), grant from the Practical Research Project for Rare and Intractable diseases from the Japan Agency for Medical Research and Development (AMED) (15ek0109082h0001, 16ek0109082h0002), and grant from the Clinical Research Project for Rare and Intractable Diseases from the Japan AMED (15lk0103018h0002, 16lk0103018h0003). The funders had no role in study design, data collection and analysis, decision to publish, or preparation of the manuscript.

**Competing interests:** The authors have declared that no competing interests exist.

## Introduction

Epilepsy affects over 70 million people worldwide[1], leading to adverse social, behavioral, health, and economic consequences. Although written records of epilepsy date back to 4000 BC, its pathophysiology remains incompletely understood[1]. As the complex mechanisms underlying epileptogenesis cannot be fully elucidated through human clinical studies, appropriate animal models are necessary.

Microphthalmia-associated transcription factor *(Mitf)*-M is expressed solely in neural crest-derived melanocytes[2]. *Mitf*-M–expressing cells are primarily found in the skin and hair follicles but also occur in other tissues, including the eyes, heart, meninges, and other brain tissues[3–6].

Skin-derived melanocytes offer a model system to investigate normal and pathological features of less accessible neurons because of their common origin and many similar signaling molecules and pathways[7]. Neurocutaneous syndromes, such as tuberous sclerosis complex (TSC), exhibit considerable overlap of dermatologic and neurologic manifestations, including epilepsy. In TSC, the prevalence of epilepsy is approximately 78%[8].

In our previous investigations of the mechanisms of melanogenesis, we constructed melanocyte-lineage *Tsc2* (a pathogenic gene in TSC) knockout mice in which *Cre* recombinase was placed under the control of regulatory elements from the *Mitf*-M gene[9]. These mice presented with the anticipated skin hypopigmentation and unexpectedly developed spontaneous neural epileptic activity as well. In the current study, we confirmed hyperactivation of the mammalian target of rapamycin (mTOR) signaling pathway, abnormal neuronal excitability, and hyperproliferation of neuronal mitochondria in the brain of these animals. We herein suggest that this may be a useful mouse model for epilepsy research, providing novel insights into the mechanisms of seizure disorders.

## Materials and methods

### Animals

*Tsc2*^flox/flox mice and *Mitf-M-Cre* mice were generated as described previously[9]. Melanocyte-specific *Tsc2* knockout mice were generated by breeding *Tsc2*^flox/flox mice with *Mitf-M-Cre* mice. Both lines maintained a C57BL/6 inbred background. The controls were littermates, either without cre or in a few cases, *Mitf-M*-cre;*Tsc2*^flox/-. For rapamycin treatment, sirolimus was purchased from Selleck (Osaka, Japan) and dissolved in distilled water for oral administration at 2.285 mg/kg/day for 3 weeks (n = 5 mice/goup). All animal experiments were conducted in accordance with the Guiding Principles for the Care and Use of Laboratory Animals, and the experimental protocol used in this study was approved by the Committee for Animal Experiments at Osaka University (Osaka, Japan).

### EEG/EMG recording

*Tsc2*^{Mitf-M} cKO mice (male, 6 weeks old at the time of surgery) were instrumented with chronically implanted EEG/EMG electrodes according to previously published procedures[10]. Briefly, a preamplifier (#8202) was surgically implanted in mice under isoflurane anesthesia. Mice (n = 3 mice/goup) were allown to recover from surgery for at least 12 h before recording was initiated. EEG/EMG data were recorded for a 24 h period using data acquisition system (#8200-K1-SE) and Sirenia Software (both from Neuroscience, inc).

### Histology and immunohistochemistry analyses

Brain tissue samples (n = 5 mice/goup) were fixed in 10% formaldehyde and embedded in paraffin. Subsequently, 4-µm sections were either stained with hematoxylin for morphological

examination or used for immunohistochemistry analysis. The following antibodies were used for immunohistochemistry: p-S6 (#4858, Cell Signaling Technology, Tokyo, Japan) at 1:100, c-FOS (ab208942, Abcam, Cambridge, UK) at 1:200, Parvalbumin (SAB4200545, Sigma) at 1:100, CaMKII-α (#11945, Cell Signaling Technology) at 1:100, COXIV (#459600, Invitrogen) at 1:200, GFAP (#12389, Cell Signaling Technology) at 1:100, and MAP2 (ab5392, Abcam) at 1:2000. The stained proteins were visualized using a Biozero confocal microscope (Keyence Co., Osaka, Japan).

## Timm staining

For Timm staining, we intracardially perfused the mice (n = 5 mice/goup) with ice-cold 1% (w/v) sodium sulfide, followed by 4% paraformaldehyde. After removal from the body, the brain was post-fixed in 10% formaldehyde overnight and embedded in paraffin. We then created 10-μm thick sagittal sections and performed modified Timm staining, as previously described[11].

## Electron microscopy examination

After dissection of the mouse brain (n = 5 mice/goup), hippocampal slices were prepared using a slicer (Narishige, ST-10, Tokyo, Japan), as previously described[12]. The slices were fixed with 2.5% glutaraldehyde in 0.1 M cacodylate buffer containing calcium chloride (pH 7.4) for 2 h and then washed three times with deionized distilled water. The samples were post-fixed in 1% $OsO_4$ in phosphate-buffered saline for 1 h, and then dehydrated with a graded ethanol series and embedded in EPON. Ultra-thin sections (70–80 nm) were cut horizontally to the bottom of the dish, transferred to grids, dual-stained with uranyl acetate and lead citrate, and observed using a Hitachi H-7650 transmission electron microscope (Hitachi, Tokyo, Japan).

## Primary culture of hippocampal pyramidal cells from adult mice

Primary neuronal cells were obtained from the hippocampus of 4-week-old wild-type and mutant mice (n = 5 mice/goup) as reported previously[13]. Briefly, the hippocampus was dissected and sliced into 0.5-mm sections in 2 mL HABG medium (40ml HA(Hibernate$^{TM}$-A Medium, Invitrogen, #A1247501; 0.8ml B27, Invitrogen, #17504; 0.1ml L-Glutamine, Invitrogen, #25030081)) at 4˚C in a 35-mm-diameter dish using tissue slicer (Dosaka microslicer, Kyoto, Japan), removing the dentate gyrus to eliminate granule cells. The sections were digested with papain (2 mg/mL, Worthington, #LS003119 in HA-Ca, BrainBits LLC) at 30˚C for 30 min. Cells were released by gentle trituration with a Pasteur pipette. Finally, primary neurons were separated using density-gradient centrifugation (OptiPrep, AXS, #1114542, XX). Cells were cultured in NeurobasalA/B27 medium (Invitrogen, #10888022 and #17504044) with L-Gin (Invitrogen, #25030149), growth factors (5 ng/mL mouse FGF2, Invitrogen, #PMG0034; 5 ng/mL mouse PDGF-BB, Invitrogen, #PMG0044), and gentamycin (Wako, #078–06061) for 1 week before the experiments.

To evaluate neuronal activity and mitochondrial quantity, cells were fixed with 4% paraformaldehyde for 20 min and then processed for the detection of neuronal antigens. The primary antibody was MAP2 (ab5392, Abcam) at 1:2000.

## Measurement of [Ca$^{2+}$]$_i$

[Ca$^{2+}$]$_i$ in single cells was detected on the basis of fura-2 fluorescence intensity, as reported previuosly[14]. Briefly, neurons grown on coverslips were rinsed twice with artificial

cerebrospinal fluid (ACSF; 127 mM NaCl, 1.5 mM KCl, 26 mM NaHCO$_3$, 1.24 mM KH$_2$PO$_4$, 10 mM glucose, 1.4 mM MgSO$_4$, 2.4 mM CaCl$_2$; SIGMA) and incubated at 37°C for 45 min in the presence of fura-2 AM (fura-2 acetoxymethyl ester, DOJINDO, #CS23) with 1.25 mmol/L probenecid (SIGMA) and 0.03% Pluronic® F-127 (SIGMA) in carbogen-bubbled ACSF. After two washes with ACSF, cells were incubated for an additional 20 min in ACSF before imaging. The coverslips were transferred to a chamber and observed by microscopy (Nikon ECLIPCE E800). The excitation wavelengths for fura-2 were 340 and 380 nm, with emission at 505 nm. For the stimulation experiments, a range of K$^+$ solutions were used: 10 mM, 30 mM, and 60 mM KCl. Fluorescence intensity was quantified using Metafluor software (Universal Imaging Corporation, West Chester, PA).

### Epilepsy behavior

cKO mice and littermates (n = 5 mice/goup) were tested for seizure behavior during the night, because mice are nocturnal animals and more active at night. Spontaneous seizure and seizures induced by the ringing of a clock every 1 hour were video-recorded for 5 days. The behaviors of the mice were scored by two independent observers, who were blinded to their genotype. In the rapamycin treatment experiments, behavior analysis was performed after 3 weeks of oral sirolimus (Selleck) in distilled water.

### Animal sacrifice

Mice were anaesthetized with a lethal dose of pentobarbital and sacrificed by intracardially perfusion using ice-cold 1% (w/v) sodium sulfide, followed by 4% paraformaldehyde. The brains were removed for primary culture of hippocampal pyramidal cells, measurement of [Ca2+]i, or post-fixed for 10% formaldehyde overnight and embedded in paraffin or cryoprotected in 30% sucrose/PBS for histologic analyses.

### Statistical analyses

Data are presented as mean ± SD. Unpaired Student's *t*-test (Microsoft Excel; Microsoft Corp., Redmond, WA) was used for comparisons between two groups. One-way ANOVA test, followed by Dunnett's post hoc test was used for multiple comparisons (Microsoft Excel). *P*-values <0.05 were considered statistically significant.

## Results

### Conditional *Tsc2* deletion caused epilepsy

We generated *Tsc2*$^{flox/flox}$;*Mitf-M-Cre* (cKO) mice by knocking out *Tsc*2 in melanocyte-lineage cells under *Mitf-M* promoter regulation[9]. Progressive recurrent epilepsy, characterized by spontaneous adduction or flexion movements of the head, trunk, limbs, and tail for 20-60s, became apparent at 4–5 weeks of age (Fig 1A; S1 Video). Almost all cKO mice appeared to have this epilepsy-like phenotype, and seizure movements were easily triggered by changing environmental status, such as ringing a clock or moving the cage suddenly. The frequency and duration of seizure-like episodes increased with age.

To further characterize these episodes, electrocorticographic activity was recorded for 6–12 hours using a digital video-EEG/EMG system (Neuroscience, inc) in cKO mice and control wild-type littermates at 6 weeks of age. Control mice showed well-organized background activity (under 100-μV spikes) during awake and at rest. By contrast, frequent (2~3 times/hour) high-amplitude sharp waves (above 300-μV spikes, over 10 seconds) were observed during

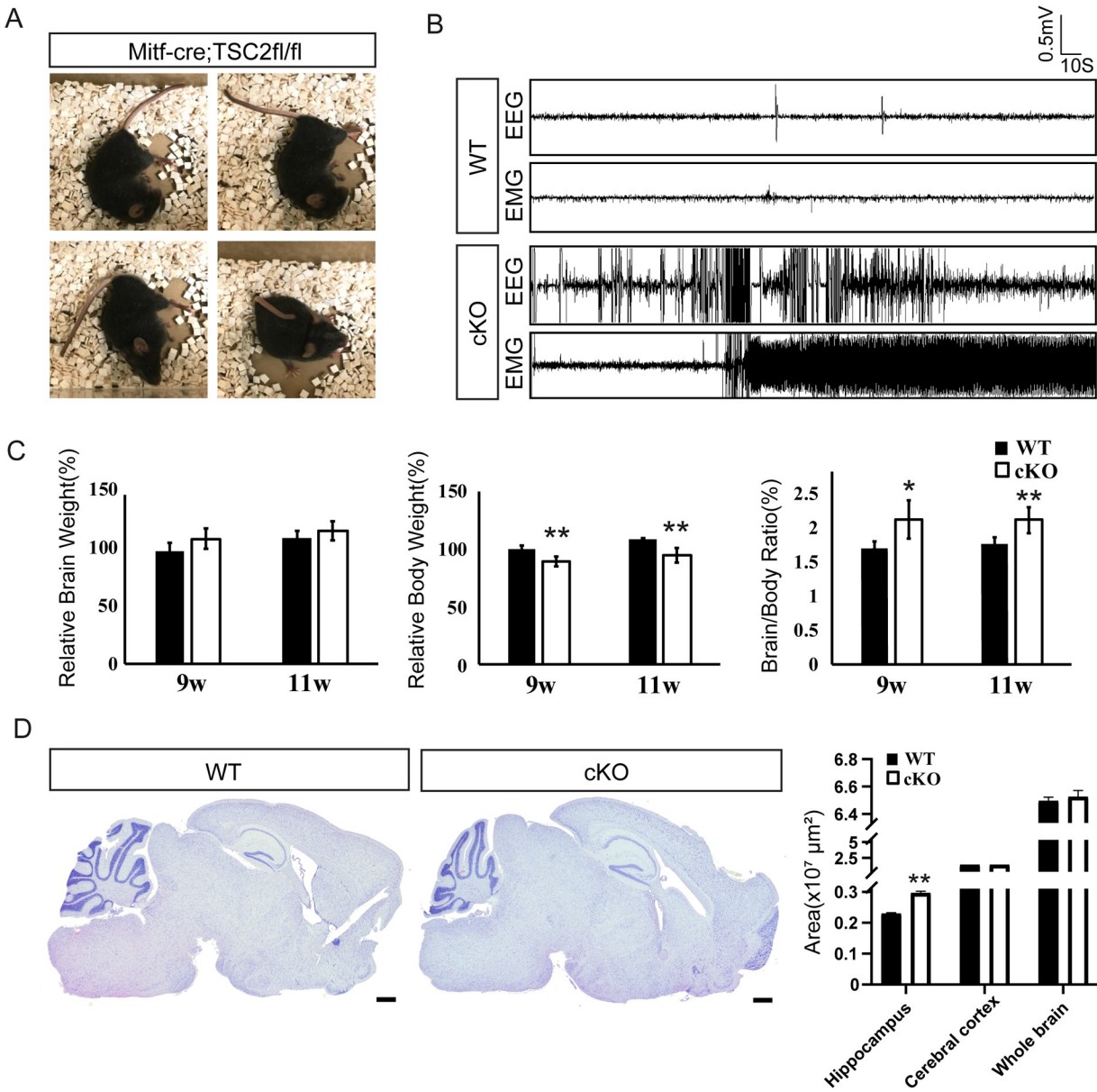

**Fig 1. Deletion of *Tsc2* resulted in epilepsy in *Tsc2^Mitf-M* cKO mice without obvious histoarchitectural changes.** A. Images captured from video recordings, showing typical spontaneous epilepsy in a 6-week-old cKO mouse. B. EEG and EMG segments (300 s) showing normal electrography in a control (WT) mouse and typical electrographic epilepsy in a cKO mouse. C. Relative brain and body weight in cKO mice compared with control (WT) mice at 9 and 11 weeks of age. $^*p < 0.05$ and $^{**}p < 0.01$ versus WT mice, n = 5 in each group, unpaired *t*-test. D. Hematoxylin staining of murine brain tissue sections, Scale bars: 600 μm. Sizes of hippocampus, Cerebral cortex and Whole brain are shown in the right panel, $^{**}p < 0.01$ versus WT mice, n = 5 in each group, unpaired *t*-test. Data in C and D are expressed as mean ± SD.

awake in the cKO mice, it was accompanied with seizure-like convulsive movements determined by video recording (Fig 1B).

Macrocephaly has been previously reported in other neuronal cell–lineage cKO mice characterized by hyperactivation of the mTOR signaling pathway in neurons[15–20]. The macrocephaly was attributed to neuronal hypertrophy secondary to an autonomous increase in the nuclear and soma size of mTOR-hyperactivated neurons[15–20]. Interestingly, body weight decreased in cKO mice, and the relative brain weight (ratio of brain weight versus body

weight) increased in cKO mice, indicated another form of macrocephaly (Fig 1C). Furthermore, hematoxylin staining revealed no aberrant histological features in the whole brains of cKO mice, however, the hippocampus increased in size compared with the control littermates (Fig 1D).

## Hyperactivation of mTOR induced neural excitation in *Tsc2*<sup>*Mitf-M*</sup> cKO mice

As a negative regulator of mTOR signaling, loss of TSC2 would be predicted to constitutive activation of mTOR [21] and subsequently phosphorylation of ribosomal protein S6 (pS6). Usually, p-S6 is regarded as a specific indicator of TSC2 loss and mTOR activation[22]. Marked hyperactivation of mTOR signaling was observed in the hippocampus, cerebral cortex, and thalamus of cKO mice, compared with control littermates (Fig 2A, panel a and b). Control mice exhibited little or no expression of pS6, especially in the hippocampus, whereas cKO mice exhibited a dramatic increase in pS6 in almost all of the hippocampus, from the dentate gyrus to the CA1 zone. To verify neuronal activity in cKO mice, immunohistochemistry staining was performed with anti-cFOS, a marker of neuronal excitability[23]. Similar to the mTOR activity (pS6) results, higher expression levels of cFOS were observed in the hippocampus, cerebral cortex, and thalamus of cKO mice, compared with control littermates (Fig 2A, panels c and d); the increase was most pronounced in the hippocampus and cerebral cortex. It could be possible that mTOR hyperactivation is inducing neuronal excitability.

To quantify expression levels of pS6 and cFOS in the hippocampus, cerebral cortex, and thalamus, we counted the cFOS- and pS6-positive cells in these regions (Fig 2B). Almost 20% of hippocampal neurons exhibited increased mTOR activity (pS6 expression) in CKO mice (Fig 2B). Furthermore, excitability (cFOS expression) of hippocampal neuronal cells, increased dramatically from 9.7% in control mice to 69% in cKO mice (Fig 2B). The cerebral cortex exhibited lower mTOR activity than the hippocampus in cKO mice, and only slightly increased neuronal excitability compared with control mice. The thalamus in cKO mice exhibited a slight increase in mTOR activity but almost no change in neuronal excitability compared with control littermates. These data suggest that the neuronal abnormality of the hippocampus may be associated with the onset of the epilepsy phenotype in this mouse model.

## Histologic changes were not observed in the hippocampal region of *Tsc2*<sup>**Mitf-M**</sup> cKO mice

A recent report indicated that the mTOR pathway regulates excitability of the hippocampal network through controlling the excitatory/inhibitory synaptic balance[24]. Therefore, we used immunofluorescence staining to examine excitatory neurons (using anti-CaMKII-a antibody) and inhibitory neurons (using anti-Parvalbumin antibody) in the mouse hippocampus (Fig 3A). Positive cells were counted, and the ratio of excitatory to inhibitory neurons was calculated (Fig 3B). No significant difference was observed in the excitatory/inhibitory synaptic balance between cKO mice and control littermates (Fig 3B). It suggests there might have some other players involved in seizure initiation and propagation, e.g. different interneuron subpopulations [25–27].

In 1989, Sutula et al. reported reorganization of mossy fiber axons, which projected abnormally into the dentate inner molecular layer in epilepsy[28]. This phenomenon, called mossy fiber sprouting, also appeared in a granule cell–lineage mTOR hyperactivation mouse model, suggesting that epilepsy might be associated with mTOR hyperactivation-induced neuronal restructuring[29]. As increased mTOR signaling was confirmed in the dentate gyrus of our *Tsc2*<sup>*Mitf-M*</sup> cKO mice (Fig 2A, panel a), we further examined the status of mossy fibers in our

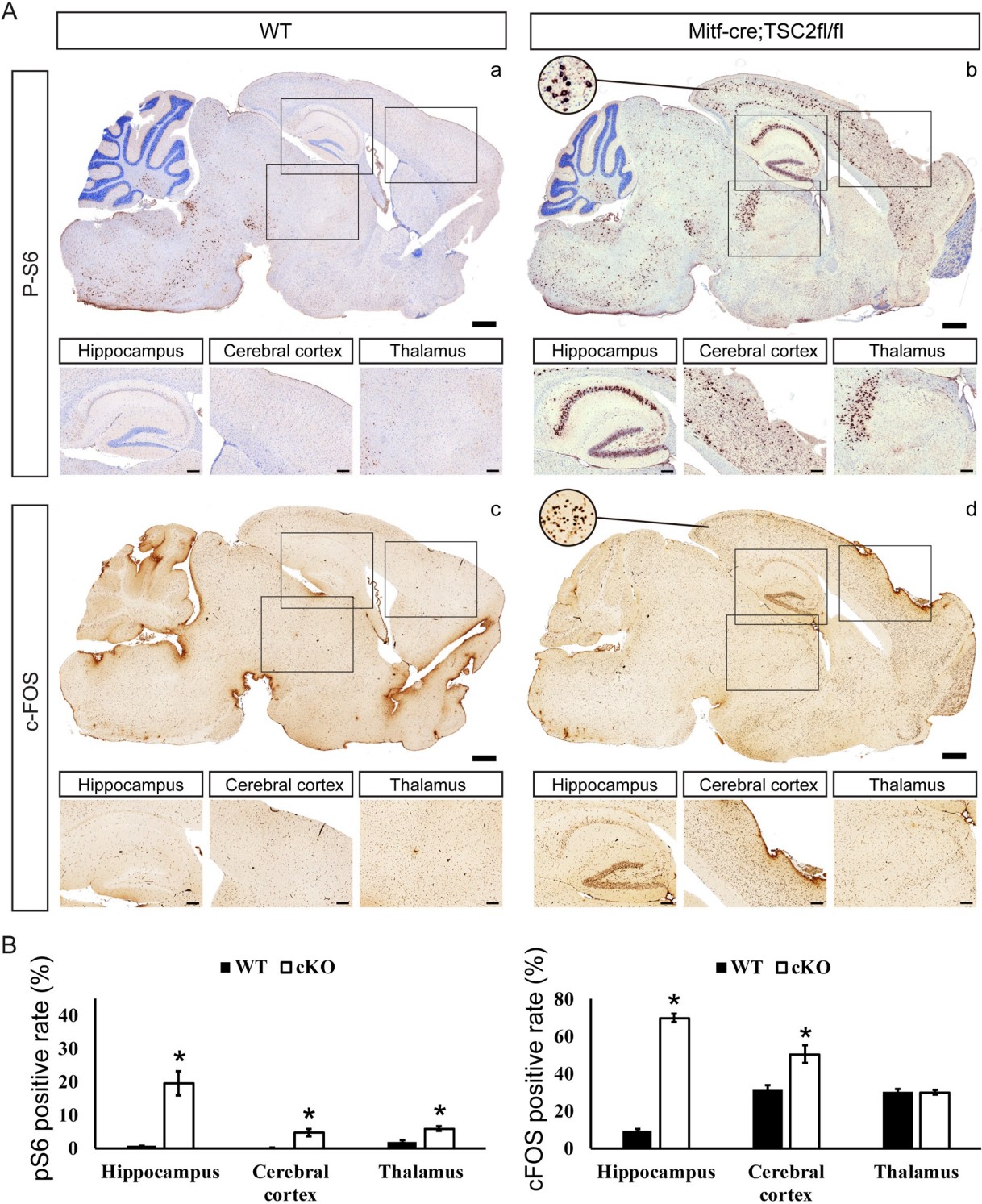

**Fig 2. Hyperactivation of mTOR induced neural excitation in $Tsc2^{Mitf-M}$ cKO mice.** A. Histoimmunostaining of whole brain sagittal sections from control (WT) mice (left panels) and cKO mice (right panels) at 5 weeks of age. p-S6 (upper panels) and c-FOS (bottom panels). The black rectangle outlines the area of hippocampus, cerebral cortex, and thalamus, and the detail is shown in the corresponding bottom panels. The circle shows representative p-S6 cytoplasmic and c-FOS nuclear positive staining. Scale bars: large bars, 600 μm; smaller bars, 200 μm. B. p-S6 and c-FOS positive rates (p-S6 or c-Fos-positive neuron cells versus all neuron cells) in the hippocampus, cerebral cortex, and thalamus. Data in C and D are expressed as mean ± SD. *$P<0.05$ versus WT mice, n = 5 in each group, unpaired $t$-test.

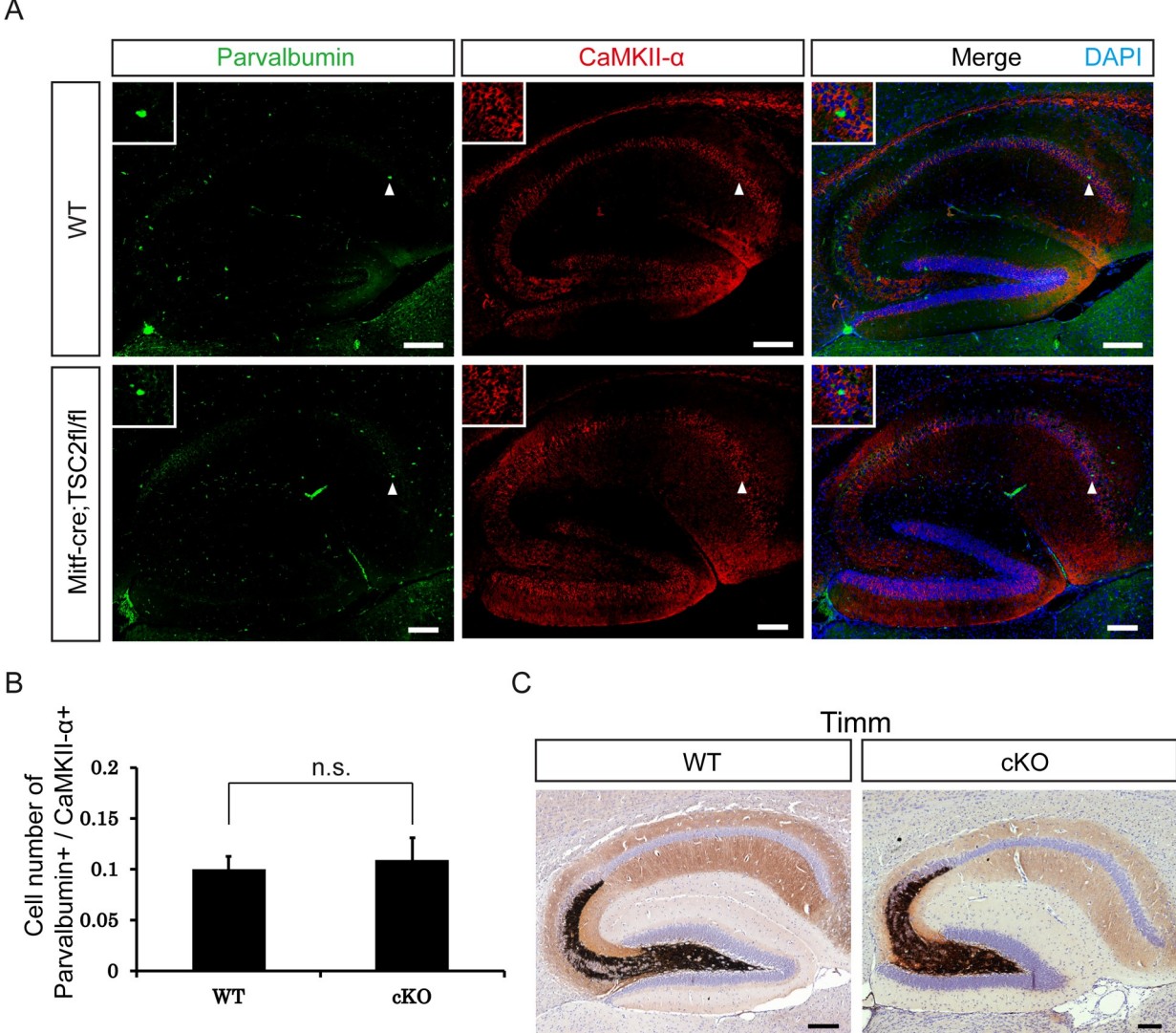

**Fig 3. Histopathological analyses of the hippocampal region in *Tsc2*^Mitf CKO mice.** A. Immunofluorescence staining showing excitatory (CaMKII-α) and inhibitory (Parvalbumin) neurons in the hippocampus. The insets show higher magnification of positive cells (arrowheads). B. Numbers of CaMKII-α-positive cells and Parvalbumin-positive cells were double-blind counted in 10 random fields per tissue section. Ratio of inhibitory to excitatory neurons were calculated (n = 5 mice). Data in B are expressed as mean ± SD. n.s. means no significance versus WT mice, unpaired *t*-test. C. Timm staining. The amount of mossy fiber sprouting is similar in cKO and control (WT) mice. Scale bars: A, 200 μm; C, 200 μm. n.s., not significant.

cKO mice by Timm staining (Fig 3C). The mossy fiber tract was of normal thickness and exhibited no sprouting in cKO mice, compared with control littermates (Fig 3C). Together, these data suggest that restructuring of neuronal pathways, excitatory/inhibitory synaptic imbalance, and mossy fiber sprouting are not involved in the mechanism of epilepsy development in our cKO mice.

## Deletion of *Tsc2* induces mitochondrial hyperproliferation in the neurons of *Tsc2*^Mitf-M cKO mice

In neuronal cell-lineage transgenic mice, mTOR hyperactivity has been previously shown to induce many structural abnormalities associated with recurrent circuit formation, including

hypertrophy of soma and dendrites, aberrant basal dendrites, impaired polarization, and enlarged axon tracts[20, 29, 30]. In the present study, we further investigated the ultrastructure of p-S6 high-expressed hippocampal pyramidal cells in the CA1 zone by electron microscopy (Fig 4A). Enlarged cell bodies and mitochondria were observed in the neurons of cKO mice. Somatic hypertrophy was confirmed in pyramidal cells of cKO mice (Fig 4A, upper panel). Furthermore, dramatic enlargement and hyperproliferation of mitochondria were observed in the pyramidal neurons of cKO mice (Fig 4A). Mitochondria increased nearly 5-fold in number and 2-fold in size, compared with neurons from control littermates (Fig 4B). To further confirm the increase in mitochondria, murine hippocampal tissue sections were assessed by histoimmunofluorescence staining with anti-COXIV antibody, a mitochondrial marker (Fig 4C). The results showed a substantial increase in the number of mitochondria, which corresponded to increased mTOR activity (anti-p-S6 binding), especially in the hippocampal CA1 region of cKO mice (Fig 4C).

## Neurons from $Tsc2^{Mitf\text{-}M}$ cKO mice were more excitable when stimulated

Previous studies have reported that neuronal hyperexcitability does not account for spontaneous epileptic activity with loss of $Tsc1$ (another mutated gene associated with TSC), suggesting that network restructuring plays a more important role in epileptic activity[24, 31, 32]. However, no structural abnormalities of the hippocampal network were observed in our cKO mice (Fig 2); therefore, we investigated the autonomous excitability of pyramidal cells. We isolated pyramidal cells from the hippocampal CA zone and confirmed by immunofluorescence staining that the majority of isolated cells were pyramidal cells (labeled with anti-MAP2) (Fig 5A). Neuronal excitability was examined by calcium imaging with the calcium-sensitive dye fura-2 (Fig 5B). Cells were depolarized by KCl, a treatment that promotes calcium influx via voltage-gated calcium channels[33]. The response to KCl was much greater in cKO pyramidal cells (Fig 5B, No.1–10) than in control pyramidal cells, and some cKO cells demonstrated slow recovery of the calcium transient (Fig 5B, No.1–2). The explanation for these findings is unclear, but they suggest that calcium dynamics in the pyramidal cells of $Tsc2^{Mitf\text{-}M}$ cKO mice are altered.

## Rapamycin revealed treatment effects

Finally, we assessed whether rapamycin treatment, which blocks the effects of mTOR, could prevent the development of abnormalities observed in our melanocyte-lineage mTOR hyperactivation mice. After 3 weeks of oral rapamycin, the epilepsy phenotype was dramatically improved both in frequency and duration of seizures (Fig 6A). Examination of mTOR activity (p-S6), neuronal excitability (c-FOS), and mitochondria (COXIV) demonstrated that rapamycin treatment downregulated mTOR signaling and promoted normalization of mitochondrial number and neuronal excitability (Fig 6B). The fluorescence intensity was quantified by ImageJ within the range of threshold limit [34] and showed same changes (Fig 6C).

## Discussion

In the present study, we found that mTOR hyperactivation resulting from loss of $Tsc2$ in $Mitf\text{-}M$-derived cell lineage was associated with the development of typical epilepsy, mitochondria hyperproliferation, and aberrant intracellular calcium dynamics. Our results showed a primary defect of mitochondrial biogenesis in pyramidal neurons, leading to dramatically enhanced sensitivity and aberrant synchronization, which may be involved in increased hippocampal network excitability.

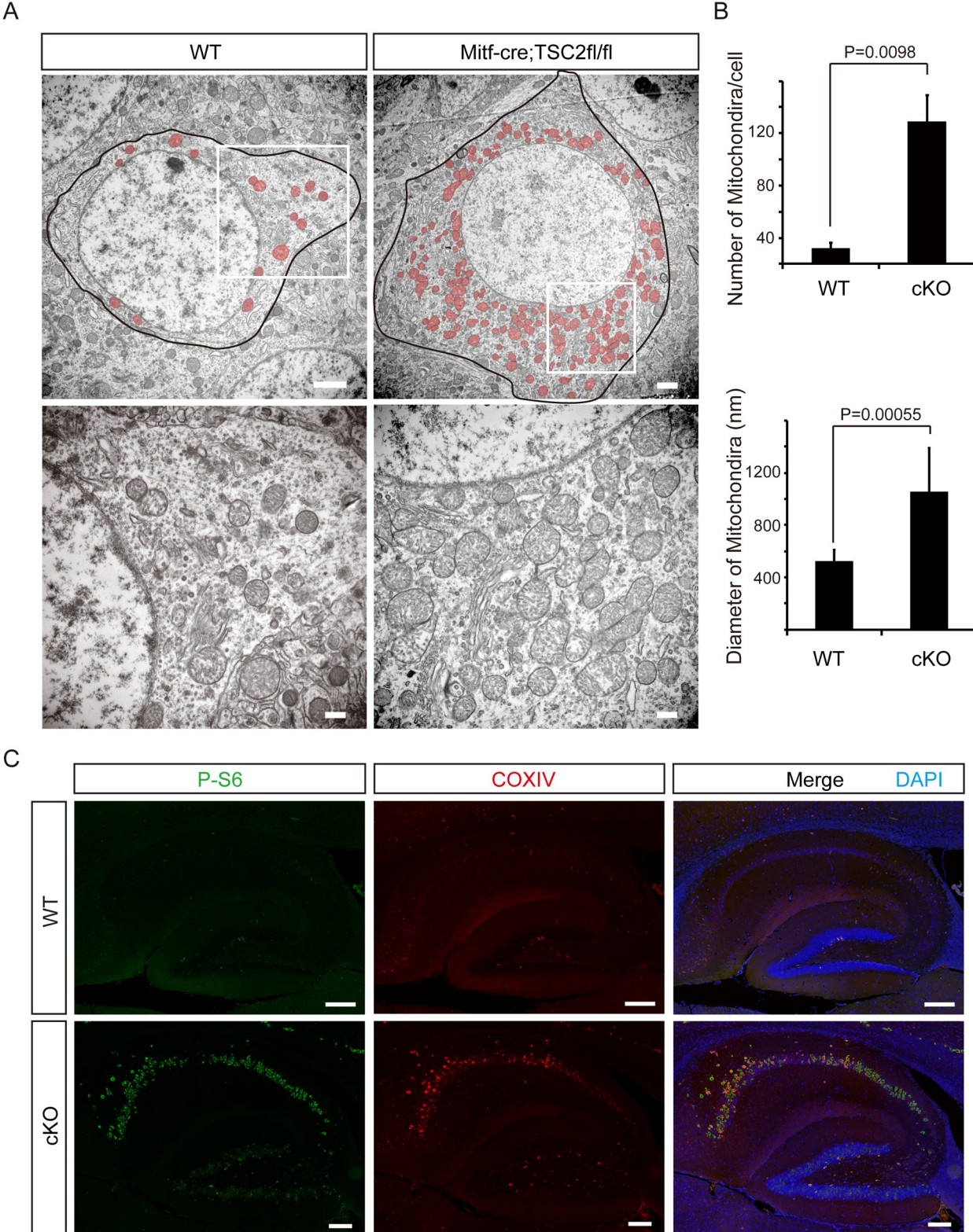

**Fig 4. Hyperproliferation of mitochondria in the neurons of *Tsc2*<sup>Mitf</sup> CKO mice.** A. Morphologic examination of p-S6 high-expressed hippocampal CA1 pyramidal cells by electron microscopy. Enlarged cell bodies and mitochondria were observed in the neurons of cKO mice. Bottom panels represent high-magnification images of the regions designated by squares. B. Quantification of mitochondria. The number of mitochondria increased more than 5-fold and the mitochondrial size increased more than 2-fold in neurons from cKO mice, compared with

control (WT) mice. (n = 20 cells/mouse, 3 mice in each group). Data in B are expressed as mean ± SD, unpaired *t*-test versus WT mice. C. Immunofluorescence staining showed hyperactivation of mTOR (p-S6) with hyperproliferation of mitochondria (COXIV) in the hippocampus of cKO mice. Scale bars: A upper panel, 2 μm; A bottom panel, 500 nm; C, 200 μm.

mTOR signaling activates the transcription of genes for mitochondrial biogenesis, including the well-known master regulator, peroxisome-proliferator-activated receptor coactivator-1α (PGC-1α)[35]. Also, mTOR modulates mitochondrial activity by enhancing interaction between transcription factor yin-yang 1 and PGC-1α [36] or by directly modifying the expression of mitochondrial proteins[37–39]. Recently, mTOR has been found to control mitochondrial activity and biogenesis through 4E-BP-dependent translational regulation[40].

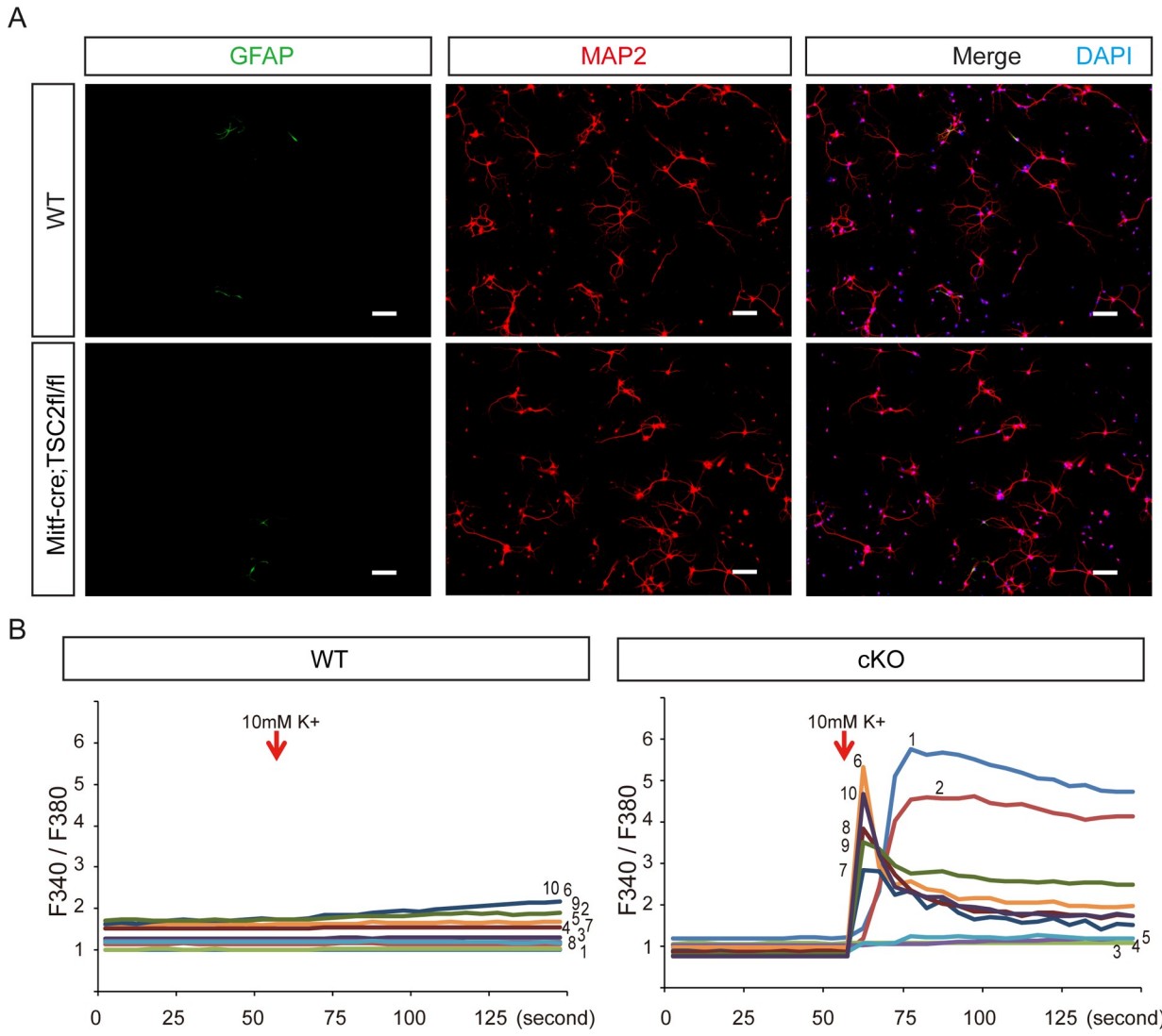

**Fig 5. Neurons from *Tsc2*$^{Mitf\,-M}$ cKO mice were more excitable than neurons from control mice.** A. Neurons were isolated from 4-week-old mice and cultured for 1 week. Immunofluorescence staining indicates that more than 80% of the isolated cells were neurons (GFAP, astrocytes; MAP2, neurons). B. Calcium imaging of cultured neurons, with corresponding traces shown at the bottom. Neurons from cKO mice respond to particularly low (10 mM) K$^+$ stimulation. Scale bars: A, 100 μm; B, 100 μm. WT, wild-type.

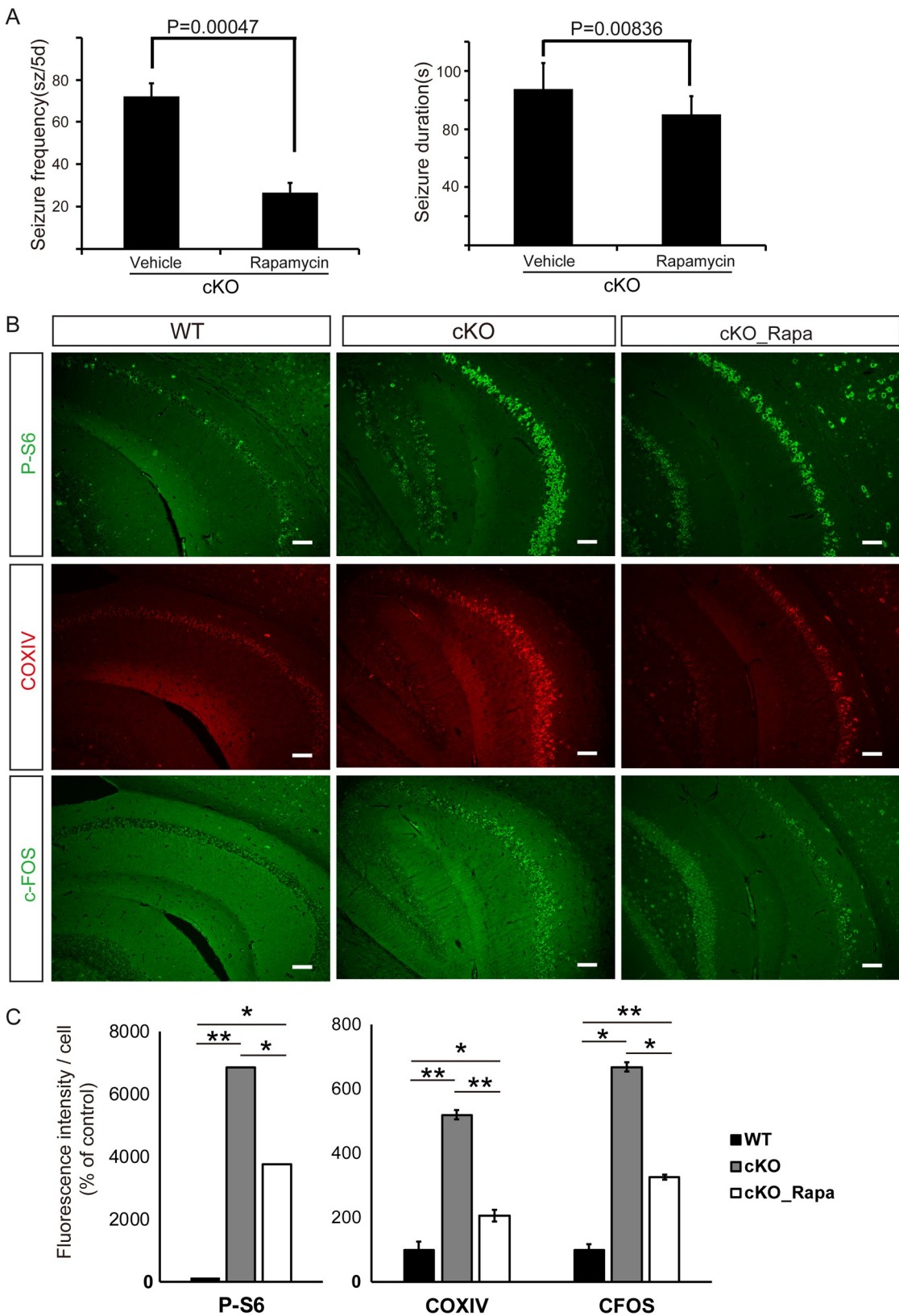

**Fig 6. Rapamycin treatment reduced seizures and number of mitochondria.** A. Frequency and duration of seizures (sz) in cKO mice in the absence or presence of rapamycin, n = 5 in each group. Data in A are expressed as mean ± SD. Unpaired *t*-test versus

vehicle-treated mice. B. Histoimmunostaining analyses of mTOR signaling (p-S6), mitochondria (COXIV), and neuronal excitation (c-FOS). Scale bars: 100 μm. WT, wild-type. C. The fluorescence intensity was quantified by ImageJ. n = 5 in each group. Data in C are expressed as mean ± SD. One-way ANOVA test, followed by Dunnett's post hoc test for multiple comparisons (WT mice versus cKO mice; cKO mice versus Rapamycin-treated cKO mice (cKO_Rapa); WT mice versus cKO_Rapa mice) was performed and adjusted $P$ values were calculated. $^*p < 0.05$, $^{**}p < 0.01$.

In our previous study[9], we detected swollen mitochondria in melanocytes in the presence of *Tsc2* deletion. In the present study, we observed not only swelling but also hyperproliferation of mitochondria in neurons.

It has been previously reported that activation of mTOR suppresses local translation of the potassium channel Kv1.1[41], resulting in increased burst firing in neurons[42, 43]. Furthermore, a direct correlation has been demonstrated between seizures and Kv1.1 gene expression [44–46]. mTOR also regulates components of neuronal RNA granules called specific RNA-binding proteins, such as fragile X mental retardation protein[47], which are involved in dendritic mRNA localization. Another line of research has demonstrated the regulatory effects of mTOR in the synthesis of new proteins in dendrites, such as PSD95 and calcium/calmodulin-dependent protein kinase[48, 49]. In the present study, we found that cKO neurons are more sensitive to potassium stimulation than controls, which may be attributed to an aberrant potassium channel or abnormal cellular ion levels.

Mitochondria contribute to various cellular processes, including ATP production, intracellular calcium signaling, cell growth and differentiation, and generation of reactive oxygen species. Neurons are critically dependent on mitochondrial function to establish membrane excitability through neurotransmission and plasticity. Electrical activity of neurons is associated with calcium influx into the cells via calcium channels, such as voltage-operated channels, store-operated channels, receptor-operated channels, and non-selective cation channels. Intracellular accumulation of calcium stimulates $Na^+/Ca^{2+}$ exchange, which maintain ionic gradients to sustain neuronal excitability[50]. The dramatic increase in number of mitochondria observed in the present study may alter intracellular $Ca^{2+}$ homeostasis, inducing hyperexcitability of neurons.

It is currently thought that a number of key molecular signaling cascades are involved in the hyperexcitability of brain tissue because controlled blocking of "master regulators" of these pathways may retard or even stop the epileptogenic process[51]. Candidate regulators that have emerged in recent years include mTOR, as well as FosB[52], p-ERK1/2[53], tropomyosin-related kinase B, brain-derived neurotrophic factor, $Zn^{2+}$-dependent cascades, and neuron-restrictive silencer factor/repressor element 1-silencing transcription factor pathways[51]. Aberrant mTOR pathway signaling has been extensively characterized in genetically-determined epilepsy in patients with mutations in the *Tsc1/2* genes in the context of TSC. This condition manifests primarily as highly-differentiated tumors or malformations in many different organs and epilepsy[15–20]. Because *Tsc* genes are negative regulators of mTOR, hyperactivation of the mTOR pathway because of *Tsc* gene mutations provides a rational mechanistic basis for abnormal cell growth and proliferation, causing tumors and other developmental lesions in TSC. Numerous transgenic mouse models of TSC have been developed by spontaneous or induced inactivation of the *Tsc1* or *Tsc2* genes in the neuronal cell lineage, which exhibit varying degrees of pathological brain abnormalities and evidence of neuronal hyperexcitability or seizures[15–20].

Physiological and anatomical studies have produced conflicting findings regarding mTOR hyperactivation-induced neurological abnormalities. For example, some research reported reduced dendritic spine density after *Tsc* deletion[54], whereas another study indicated that the density increased[20]. As *Tsc* deletion produces a neurodevelopmental disorder, these

discrepancies may depend on the role of mTOR in different types of cells and at different stages of development.

Until now, rodent models of spontaneous recurrent epilepsy have been generated by chemoconvulsants (primarily pilocarpine and kainic acid), neonatal hypoxia, traumatic brain injuries, electrical stimulation or genetic manipulations [55]. However, none of these models provide cell-type specificity in the brain[55]. By contrast, our mouse model involves *Tsc* knockout in specific *Mitf-M*-lineage cells.

In previous research regarding mTOR-associated epilepsy, structural abnormalities of neurons have been considered the primary etiologic factor. In the present study, we generated a novel epilepsy mouse model based on mTOR hyperactivation. The model is characterized by abnormal mitochondria, which may be responsible for the development of epilepsy by directly upregulating neuronal excitability. This model may be used to facilitate the development of new therapeutic interventions for seizure disorders.

## Supporting information

**S1 Video. Video recording of typical epilepsy.**
(MP4)

## Acknowledgments

We thank Professor Akitsugu Yamamoto (Faculty of Bioscience, Nagahama Institute of Bioscience and Technology), Nishida Kenju (Department of Dermatology, Course of Integrated Medicine, Graduate School of Medicine, Osaka University), Oiki Eiji (Center for Medical Research and Education, Graduate School of Medicine, Osaka University), and Neuroscience co.ltd (Japan) for their expert technical assistance. We also thank Takahumi Kawai, Tomomitsu Miyoshi, and Professor Yasushi Okamura (Department of Physiology, Course of Integrated Medicine, Graduate School of Medicine, Osaka University) for their kind advice.

## Author Contributions

**Conceptualization:** Fei Yang, Lingli Yang, Mari Wataya-Kaneda, Lanting Teng, Ichiro Katayama.

**Data curation:** Fei Yang, Lingli Yang.

**Formal analysis:** Fei Yang, Lingli Yang, Mari Wataya-Kaneda.

**Funding acquisition:** Mari Wataya-Kaneda.

**Investigation:** Fei Yang, Lingli Yang, Mari Wataya-Kaneda, Lanting Teng.

**Methodology:** Fei Yang, Lingli Yang, Lanting Teng.

**Project administration:** Fei Yang, Lingli Yang, Mari Wataya-Kaneda.

**Resources:** Fei Yang, Lingli Yang.

**Software:** Fei Yang, Lingli Yang.

**Supervision:** Mari Wataya-Kaneda, Ichiro Katayama.

**Validation:** Fei Yang, Lingli Yang, Mari Wataya-Kaneda, Lanting Teng, Ichiro Katayama.

**Visualization:** Fei Yang, Lingli Yang, Mari Wataya-Kaneda, Lanting Teng.

**Writing – original draft:** Fei Yang, Lingli Yang.

**Writing – review & editing:** Fei Yang, Lingli Yang, Mari Wataya-Kaneda, Lanting Teng, Ichiro Katayama.

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
