## [Decision Letter · Decision Letter 0]

22 Oct 2019

PONE-D-19-25631

Epilepsy in a melanocyte-lineage mTOR hyperactivation mouse model: a novel epilepsy model

PLOS ONE

Dear Dr. Wataya-Kaneda,

Thank you for submitting your manuscript to PLOS ONE. After careful consideration, we feel that it has merit but does not fully meet PLOS ONE’s publication criteria as it currently stands. Therefore, we invite you to submit a revised version of the manuscript that addresses the points raised during the review process.

We would appreciate receiving your revised manuscript by Dec 06 2019 11:59PM. To enhance the reproducibility of your results, we recommend that if applicable you deposit your laboratory protocols in protocols.io, where a protocol can be assigned its own identifier (DOI) such that it can be cited independently in the future. For instructions see: http://journals.plos.org/plosone/s/submission-guidelines#loc-laboratory-protocols

We look forward to receiving your revised manuscript.

Kind regards,

Giuseppe Biagini, MD

Academic Editor

PLOS ONE

**Journal Requirements:**

3. To comply with PLOS ONE submissions requirements, please provide methods of sacrifice in the Methods section of your manuscript.

**Comments to the Author**

1. Is the manuscript technically sound, and do the data support the conclusions?

Reviewer #1: Yes

Reviewer #2: Yes

2. Has the statistical analysis been performed appropriately and rigorously? 

Reviewer #1: Yes

Reviewer #2: I Don't Know

3. Have the authors made all data underlying the findings in their manuscript fully available?

Reviewer #1: Yes

Reviewer #2: Yes

4. Is the manuscript presented in an intelligible fashion and written in standard English?

Reviewer #1: Yes

Reviewer #2: Yes

5. Review Comments to the Author

Reviewer #1: 1) In material and methods, you should precise the total number of animals used in your experiment.

2) “The fluorescence intensity was quantified by ImageJ and showed same changes (Fig. 6C).” Could please explain the method of quantification by ImageJ?

3) In Fig. 3A, 3C, 4A and 4C, the scale bars are different in cKO vs WT. Why?

4) It would be preferable to have some sentences about electrographic seizures resumed in a single paragraph. Particularly:

• you should deeply precise how their activity different from baseline activity (duration, rhythms, frequency range and amplitude)

• you should also precise how you define the onset of the seizure: shape of the first part of the seizure, occurrence …

• finally, you should explain how you characterize the end of the seizure: return to baseline activity, post-ictal depression...

5) In results, you reported that “A recent report indicated that the mTOR pathway regulates excitability of the hippocampal network through controlling the excitatory/inhibitory synaptic balance (19). Therefore, we used immunofluorescence staining to examine excitatory neurons (using anti-CaMKII-a antibody) and inhibitory neurons (using anti-Parvalbumin antibody) in the mouse hippocampus (Fig. 3A). Positive cells were counted, and the ratio of excitatory to inhibitory neurons was calculated (Fig. 3B). No significant difference was observed in the excitatory/inhibitory synaptic balance between cKO mice and control littermates (Fig. 3B).” In this regard, I think you should deeply discuss the crucial importance of different interneuron subpopulations in seizure initiation and propagation (refer to “de Curtis, M., and Avoli, M. (2016). GABAergic networks jump-start focal seizures. Epilepsia”), but also in seizure modulation (refer to "Khoshkhoo, S., Vogt, D., and Sohal, V. S. (2017). Dynamic, Cell-Type-Specific Roles for GABAergic Interneurons in a Mouse Model of Optogenetically Inducible Seizures. Neuron"). Moreover, only specific interneuron subpopulations but not others seem to be involved in the anticonvulsant effects in other models of seizures (refer to “Lucchi et al. (2017). Involvement of PPARγ in the anticonvulsant activity of EP-80317, a ghrelin receptor antagonist. Frontiers in Pharmacology”.)

6) In the discussion, the comparison between mTOR and other markers of neuronal activation used in literature should be done. For instance, you could refer to “Yutsudo et al. (2013). fosB-null mice display impaired adult hippocampal neurogenesis and spontaneous epilepsy with depressive behavior. Neuropsychopharmacology” and to “Giordano et al. (2016). Progressive seizure aggravation in the repeated 6-hz corneal stimulation model is accompanied by marked increase in hippocampal p-ERK1/2 immunoreactivity. Frontiers in Cellular Neuroscience”.

Reviewer #2: In the current Ms, the authors investigate further their melanocyte-lineage mTOR hyperactivation mouse model and try to describe it as a new model of epilepsy.

This Ms provides very important data to the field of epilepsy research. However, after reading the title of this Ms, I was quite disappointed when I realised that the authors did not try any classical antiepileptic drug on their supposedly new model of epilepsy. Has this been considered?

I have some other concerns listed below.

A) The “Materials and Methods” section lacks sufficient information.

To give few examples:

1) This is not easy to find the number of animals used for each protocol.

2) I could not find a clear description of how the animals were sacrified or anesthetized. This comment applies for almost all sections.

P5 Ln84 “Histology and immunohistochemistry analyses”: How animals are sacrificed before the brains being fixed?

P6Ln95: “Timm staining”: How were the rats anesthetized?

P6LnLn100: “Electron microscopy examination”: How were the rats sacrificed?

3) P7LnLn110: “Primary culture of hippocampal pyramidal cells from adult mice”: How were the slices cut?

4) P7Ln122: Measurement of [Ca2+]i: Please could you check the composition of your ACSF? What was the osmolarity of this solution?

B) In the discussion at LnP310: “Until now, rodent models of spontaneous recurrent epilepsy have been generated by chemoconvulsants (primarily pilocarpine and kainic acid), neonatal hypoxia, traumatic brain injuries or electrical stimulation(44).” Please note that genetic models of epilepsy are also available to study some types of epilepsy.

6. PLOS authors have the option to publish the peer review history of their article (what does this mean?). If published, this will include your full peer review and any attached files.

Reviewer #1: No

Reviewer #2: No

---

## [Author Response · Author response to Decision Letter 0]

1 Dec 2019

COMMENTS FROM REVIEWER #1: 

COMMENT: 1. In material and methods, you should precise the total number of animals used in your experiments. 

RESPONSE: Thank you for your careful reading and valuable comment. As requested, we have added the description about the number of mice we used in each experiment in Materials and methods section in red with underline.

COMMENT: 2. “The fluorescence intensity was quantified by ImageJ and showed same changes (Fig. 6C).” Could please explain the method of quantification by ImageJ?

RESPONSE: Thank you for this important comment. As suggested, we have added the description of quantification method as indicated below. 

(manuscript, Results section, page 14, line 261-262, in red with underline).

The fluorescence intensity was quantified by ImageJ within the range of threshold limit (Jensen EC. Quantitative analysis of histological staining and fluorescence using ImageJ. Anat Rec (Hoboken). 2013 Mar;296(3):378-81).

COMMENT: 3. In Fig. 3A, 3C, 4A and 4C, the scale bars are different in cKO vs WT. Why?

RESPONSE: Thank you very much for the careful reading of our manuscript. Because the hippocampus is bigger than WT (as shown in Fig 1C), in Figure 3A, 3C and 4C, to put the whole hippocampus of cKO into the same frame size to WT, images of WT and cKO were shown in different magnification with their different scales respectively. And the hippocampal CA1 pyramidal cells of cKO is also a little bit bigger than WT, in Figure 4A, to put the whole hippocampal CA1 pyramidal cell of cKO into the same frame size to WT, images of WT and cKO were shown in different magnification with their different scales respectively.

COMMENT: 4. It would be preferable to have some sentences about electrographic seizures resumed in a single paragraph. Particularly:

• you should deeply precise how their activity different from baseline activity (duration, rhythms, frequency range and amplitude)

• you should also precise how you define the onset of the seizure: shape of the first part of the seizure, occurrence …

• finally, you should explain how you characterize the end of the seizure: return to baseline activity, post-ictal depression...

RESPONSE: Thank you for your kind advice. According to your suggestion, we have modified the description as indicated below (manuscript, results section, page 9, line 160-166, in red with underline).

To further characterize these episodes, electrocorticographic activity was recorded for 6-12 hours using a digital video-EEG/EMG system (Neuroscience, inc) in cKO mice and control wild-type littermates at 6 weeks of age. Control mice showed well-organized background activity (under 100-µV spikes) during awake and at rest. By contrast, frequent (2~3 times/hour) high-amplitude sharp waves (above 300-µV spikes, over 10 seconds) were observed during awake in the cKO mice, it was accompanied with seizure-like convulsive movements determined by video recording (Fig 1B).

COMMENT: 5. In results, you reported that “A recent report indicated that the mTOR pathway regulates excitability of the hippocampal network through controlling the excitatory/inhibitory synaptic balance (19). Therefore, we used immunofluorescence staining to examine excitatory neurons (using anti-CaMKII-a antibody) and inhibitory neurons (using anti-Parvalbumin antibody) in the mouse hippocampus (Fig. 3A). Positive cells were counted, and the ratio of excitatory to inhibitory neurons was calculated (Fig. 3B). No significant difference was observed in the excitatory/inhibitory synaptic balance between cKO mice and control littermates (Fig. 3B).” In this regard, I think you should deeply discuss the crucial importance of different interneuron subpopulations in seizure initiation and propagation (refer to “de Curtis, M., and Avoli, M. (2016). GABAergic networks jump-start focal seizures. Epilepsia”), but also in seizure modulation (refer to "Khoshkhoo, S., Vogt, D., and Sohal, V. S. (2017). Dynamic, Cell-Type-Specific Roles for GABAergic Interneurons in a Mouse Model of Optogenetically Inducible Seizures. Neuron"). Moreover, only specific interneuron subpopulations but not others seem to be involved in the anticonvulsant effects in other models of seizures (refer to “Lucchi et al. (2017). Involvement of PPARγ in the anticonvulsant activity of EP-80317, a ghrelin receptor antagonist. Frontiers in Pharmacology”.) RESPONSE: Thank you for this helpful advice. According to your suggestion, we have added these three references (25-27) to our manuscript and modified the description as indicated below (manuscript, results section, page 11, line 203-211, in red with underline). “A recent report indicated that the mTOR pathway regulates excitability of the hippocampal network through controlling the excitatory/inhibitory synaptic balance(24). Therefore, we used immunofluorescence staining to examine excitatory neurons (using anti-CaMKII-a antibody) and inhibitory neurons (using anti-Parvalbumin antibody) in the mouse hippocampus (Fig 3A). Positive cells were counted, and the ratio of excitatory to inhibitory neurons was calculated (Fig 3B). No significant difference was observed in the excitatory/inhibitory synaptic balance between cKO mice and control littermates (Fig 3B). It suggests there might have some other players involved in seizure initiation and propagation, e.g. different interneuron subpopulations (25-27).”

COMMENT: 6. In the discussion, the comparison between mTOR and other markers of neuronal activation used in literature should be done. For instance, you could refer to “Yutsudo et al. (2013). fosB-null mice display impaired adult hippocampal neurogenesis and spontaneous epilepsy with depressive behavior. Neuropsychopharmacology” and to “Giordano et al. (2016). Progressive seizure aggravation in the repeated 6-hz corneal stimulation model is accompanied by marked increase in hippocampal p-ERK1/2 immunoreactivity. Frontiers in Cellular Neuroscience”.

RESPONSE: Thank you for this valuable comment. we have added these three references (52, 53) to our manuscript and modified the description as indicated below (manuscript, discussion section, page 16, line 304-307, in red with underline).

“Candidate regulators that have emerged in recent years include mTOR, as well as FosB(52), p-ERK1/2(53), tropomyosin-related kinase B, brain-derived neurotrophic factor, Zn2+-dependent cascades, and neuron-restrictive silencer factor/repressor element 1-silencing transcription factor pathways(51).” 

COMMENTS FROM REVIEWER #2: 

COMMENT: In the current Ms, the authors investigate further their melanocyte-lineage mTOR hyperactivation mouse model and try to describe it as a new model of epilepsy. This Ms provides very important data to the field of epilepsy research.

RESPONSE: Thank you for your careful reading and thank for your interest and compliments on our study. 

COMMENT: However, after reading the title of this Ms, I was quite disappointed when I realised that the authors did not try any classical antiepileptic drug on their supposedly new model of epilepsy. Has this been considered?

RESPONSE: Thank you for your kind comment. We have tried to treat our mice using the classical mTOR inhibitor-Rapamycin, after 3 weeks of oral rapamycin treatment, epilepsy phenotype was dramatically improved both in frequency and duration of seizures (Fig 6A), however, we have not treated with some other classical clinical antiepileptic drugs. We would like to investigate that in our future study and report that in our next paper.

COMMENT: I have some other concerns listed below.

A) The “Materials and Methods” section lacks sufficient information.

RESPONSE: Thank you for your careful reading and valuable comment. As requested, we have modified our manuscript according to your suggestions.

COMMENT: 1. This is not easy to find the number of animals used for each protocol.

RESPONSE: Thank you for your careful reading and valuable comment. As requested, we have added the description about the number of mice we used in each experiment in Materials and methods section in red with underline.

COMMENT: 2. I could not find a clear description of how the animals were sacrified or anesthetized. This comment applies for almost all sections. P5 Ln84 “Histology and immunohistochemistry analyses”: How animals are sacrificed before the brains being fixed? P6Ln95: “Timm staining”: How were the rats anesthetized? P6LnLn100: “Electron microscopy examination”: How were the rats sacrificed?

RESPONSE: Thank you for your important comments. As requested, we have added the description of animal sacrifice as indicated below. 

(manuscript, Methods section, page 8, line 140-144, in red with underline).

 “Mice were anaesthetized with a lethal dose of pentobarbital and sacrificed by intracardially perfusion using ice-cold 1% (w/v) sodium sulfide, followed by 4% paraformaldehyde. The brains were removed, post-fixed for 10% formaldehyde overnight and embedded in paraffin or cryoprotected in 30% sucrose/PBS.”

COMMENT: 3. P7LnLn110: “Primary culture of hippocampal pyramidal cells from adult mice”: How were the slices cut?

RESPONSE: Thank you for your important comments. As requested, we have added the description about the slice cut as indicated below. 

(manuscript, Methods section, page 7, line 106-116, in red with underline).

“Primary neuronal cells were obtained from the hippocampus of 4-week-old wild-type and mutant mice (n = 5 mice/goup) as reported previously(13). Briefly, the hippocampus was dissected and sliced into 0.5-mm sections using tissue slicer (Dosaka microslicer, Kyoto, Japan), removing the dentate gyrus to eliminate granule cells. The sections were digested with papain (2 mg/mL, Worthington, #LS003119 in HA-Ca, BrainBits LLC) at 30°C for 30 min. Cells were released by gentle trituration with a Pasteur pipette. Finally, primary neurons were separated using density-gradient centrifugation (OptiPrep, AXS, #1114542, XX). Cells were cultured in NeurobasalA/B27 medium (Invitrogen, #10888022 and #17504044) with L-Gin (Invitrogen, #25030149), growth factors (5 ng/mL mouse FGF2, Invitrogen, #PMG0034; 5 ng/mL mouse PDGF-BB, Invitrogen, #PMG0044), and gentamycin (Wako, #078-06061) for 1 week before the experiments.”

COMMENT: 4. P7Ln122: Measurement of [Ca2+]i: Please could you check the composition of your ACSF? What was the osmolarity of this solution?

RESPONSE: Thank you for your careful reading and the important comment. Regret for this mistake, as suggested, we have added the reference (14) in our manuscript and revised our mistake in the concentration of NaCl (revised 45mM to 127 mM NaCl). 

(manuscript, Methods section, page 7, line 121-126, in red with underline).

[Ca2+]i in single cells was detected on the basis of fura-2 fluorescence intensity, as reported previuosly(14). Briefly, neurons grown on coverslips were rinsed twice with artificial cerebrospinal fluid (ACSF; 127 mM NaCl, 1.5 mM KCl, 26 mM NaHCO3, 1.24 mM KH2PO4, 10 mM glucose, 1.4 mM MgSO4, 2.4 mM CaCl2; SIGMA) and incubated at 37°C for 45 min in the presence of fura-2 AM (fura-2 acetoxymethyl ester, DOJINDO, #CS23) with 1.25 mmol/L probenecid (SIGMA) and 0.03% Pluronic® F-127 (SIGMA) in ACSF.

COMMENT: B) In the discussion at LnP310: “Until now, rodent models of spontaneous recurrent epilepsy have been generated by chemoconvulsants (primarily pilocarpine and kainic acid), neonatal hypoxia, traumatic brain injuries or electrical stimulation(44).” Please note that genetic models of epilepsy are also available to study some types of epilepsy.

RESPONSE: Thank you for the important comment. According to your suggestion, we have modified the description as indicated below.

(marked manuscript, Discussion section, page 18, line 324-326, in red with underline). “Until now, rodent models of spontaneous recurrent epilepsy have been generated by chemoconvulsants (primarily pilocarpine and kainic acid), neonatal hypoxia, traumatic brain injuries, electrical stimulation or genetic manipulations (55).”

---

## [Decision Letter · Decision Letter 1]

20 Dec 2019

PONE-D-19-25631R1

Epilepsy in a melanocyte-lineage mTOR hyperactivation mouse model: a novel epilepsy model

PLOS ONE

Dear Dr. Wataya-Kaneda,

Thank you for submitting your manuscript to PLOS ONE. After careful consideration, we feel that it has merit but does not fully meet PLOS ONE’s publication criteria as it currently stands. Therefore, we invite you to submit a revised version of the manuscript that addresses the points raised during the review process.

We would appreciate receiving your revised manuscript by Feb 03 2020 11:59PM. To enhance the reproducibility of your results, we recommend that if applicable you deposit your laboratory protocols in protocols.io, where a protocol can be assigned its own identifier (DOI) such that it can be cited independently in the future. For instructions see: http://journals.plos.org/plosone/s/submission-guidelines#loc-laboratory-protocols

We look forward to receiving your revised manuscript.

Kind regards,

Giuseppe Biagini, MD

Academic Editor

PLOS ONE

Reviewers' comments:

Reviewer's Responses to Questions

**Comments to the Author**

1. If the authors have adequately addressed your comments raised in a previous round of review and you feel that this manuscript is now acceptable for publication, you may indicate that here to bypass the “Comments to the Author” section, enter your conflict of interest statement in the “Confidential to Editor” section, and submit your "Accept" recommendation.

Reviewer #1: All comments have been addressed

Reviewer #2: All comments have been addressed

2. Is the manuscript technically sound, and do the data support the conclusions?

Reviewer #1: (No Response)

Reviewer #2: Yes

3. Has the statistical analysis been performed appropriately and rigorously? 

Reviewer #1: (No Response)

Reviewer #2: I Don't Know

4. Have the authors made all data underlying the findings in their manuscript fully available?

Reviewer #1: (No Response)

Reviewer #2: Yes

5. Is the manuscript presented in an intelligible fashion and written in standard English?

Reviewer #1: (No Response)

Reviewer #2: Yes

6. Review Comments to the Author

Reviewer #1: The authors have adequately addressed my comments raised in a previous round of review and I feel that this manuscript is now acceptable for publication.

Reviewer #2: The Ms has improved following the authors's reply to the reviewers's comments. However in my opinion, supplemental information particularly in the method section, are needed to make this Ms a little bit clearer.

1) Primary culture of hippocampal pyramidal cells from adult mice: Please could you precise in which media the slices were cut?

2) Measurement of [Ca2+]i: Please could you precise if the ACSF was "bubbled" with carbogen?

3) Animal sacrifice: one can read "Mice were anaesthetized with a lethal dose of pentobarbital and sacrificed by intracardially perfusion using ice-cold 1% (w/v) sodium sulfide, followed by 4% paraformaldehyde." I am a little bit confused. Did the authors perform Primary culture of hippocampal pyramidal cells and measurement of [Ca2+]i on cells from animals that have been perfused with 4% paraformaldehyde?

7. PLOS authors have the option to publish the peer review history of their article (what does this mean?). If published, this will include your full peer review and any attached files.

Reviewer #1: No

Reviewer #2: No

---

## [Author Response · Author response to Decision Letter 1]

7 Jan 2020

COMMENTS FROM REVIEWER #1: 

“The authors have adequately addressed my comments raised in a previous round of review and I feel that this manuscript is now acceptable for publication.” 

RESPONSE: Thank you for your careful reading and valuable suggestions for revising and improving our manuscript. 

COMMENTS FROM REVIEWER #2: 

“The Ms has improved following the authors's reply to the reviewers's comments. However, in my opinion, supplemental information particularly in the method section, are needed to make this Ms a little bit clearer.”

RESPONSE: Thank you for your careful reading and important suggestions. We have revised our manuscript according to your comments and suggestions. 

COMMENT: 1. “Primary culture of hippocampal pyramidal cells from adult mice: Please could you precise in which media the slices were cut?”

RESPONSE: Thank you for your important comments. As requested, we have added the description about the medium as indicated below. 

(Manuscript with Track Changes, Methods section, page 7, line 106-111, in red with underline).

“Primary neuronal cells were obtained from the hippocampus of 4-week-old wild-type and mutant mice (n = 5 mice/goup) as reported previously(13). Briefly, the hippocampus was dissected and sliced into 0.5-mm sections in 2 mL HABG medium (40ml HA(HibernateTM-A Medium, Invitrogen, #A1247501; 0.8ml B27, Invitrogen, #17504; 0.1ml L-Glutamine, Invitrogen, #25030081)) at 4°C in a 35-mm-diameter dish using tissue slicer (Dosaka microslicer, Kyoto, Japan), removing the dentate gyrus to eliminate granule cells.”

COMMENT: 2. “Measurement of [Ca2+]i: Please could you precise if the ACSF was "bubbled" with carbogen?”

RESPONSE: Thank you for your important comments. In our experiments, ACSF has been bubbled with carbogen. As requested, we have added the description as indicated below. 

(Manuscript with Track Changes, Methods section, page 8, line 124-129, in red with underline).

“Briefly, neurons grown on coverslips were rinsed twice with artificial cerebrospinal fluid (ACSF; 127 mM NaCl, 1.5 mM KCl, 26 mM NaHCO3, 1.24 mM KH2PO4, 10 mM glucose, 1.4 mM MgSO4, 2.4 mM CaCl2; SIGMA) and incubated at 37°C for 45 min in the presence of fura-2 AM (fura-2 acetoxymethyl ester, DOJINDO, #CS23) with 1.25 mmol/L probenecid (SIGMA) and 0.03% Pluronic® F-127 (SIGMA) in carbogen-bubbled ACSF.”

COMMENT: 3. “Animal sacrifice: one can read "Mice were anaesthetized with a lethal dose of pentobarbital and sacrificed by intracardially perfusion using ice-cold 1% (w/v) sodium sulfide, followed by 4% paraformaldehyde." I am a little bit confused. Did the authors perform Primary culture of hippocampal pyramidal cells and measurement of [Ca2+]i on cells from animals that have been perfused with 4% paraformaldehyde?”

RESPONSE: Thank you for your careful reading and the important comment. We have added the information as indicated below. 

(Manuscript with Track Changes, Methods section, page 8, line 143-147, in red with underline).

“Mice were anaesthetized with a lethal dose of pentobarbital and sacrificed by intracardially perfusion using ice-cold 1% (w/v) sodium sulfide, followed by 4% paraformaldehyde. The brains were removed for primary culture of hippocampal pyramidal cells, measurement of [Ca2+]i, or post-fixed for 10% formaldehyde overnight and embedded in paraffin or cryoprotected in 30% sucrose/PBS for histologic analyses.”

---

## [Decision Letter · Decision Letter 2]

9 Jan 2020

PONE-D-19-25631R2

Epilepsy in a melanocyte-lineage mTOR hyperactivation mouse model: a novel epilepsy model

PLOS ONE

Dear Dr Wataya-Kaneda,

Thank you for submitting your manuscript to PLOS ONE. After careful consideration, we feel that it has merit but does not fully meet PLOS ONE’s publication criteria as it currently stands. Specifically, I invite you to complete and correct description of statistical analysis because:

- results of statistical analysis are not detailed in results, and should be better illustrated in legends to figures;

- in methods, I read that you used "two-way analysis of variance", but the post hoc test is not indicated.

We would appreciate receiving your revised manuscript by Feb 23 2020 11:59PM. To enhance the reproducibility of your results, we recommend that if applicable you deposit your laboratory protocols in protocols.io, where a protocol can be assigned its own identifier (DOI) such that it can be cited independently in the future. For instructions see: http://journals.plos.org/plosone/s/submission-guidelines#loc-laboratory-protocols

We look forward to receiving your revised manuscript.

Kind regards,

Giuseppe Biagini, MD

Academic Editor

PLOS ONE

Reviewers' comments:

Reviewer's Responses to Questions

**Comments to the Author**

1. If the authors have adequately addressed your comments raised in a previous round of review and you feel that this manuscript is now acceptable for publication, you may indicate that here to bypass the “Comments to the Author” section, enter your conflict of interest statement in the “Confidential to Editor” section, and submit your "Accept" recommendation.

Reviewer #2: All comments have been addressed

2. Is the manuscript technically sound, and do the data support the conclusions?

Reviewer #2: Yes

3. Has the statistical analysis been performed appropriately and rigorously? 

Reviewer #2: I Don't Know

4. Have the authors made all data underlying the findings in their manuscript fully available?

Reviewer #2: Yes

5. Is the manuscript presented in an intelligible fashion and written in standard English?

Reviewer #2: Yes

6. Review Comments to the Author

Reviewer #2: (No Response)

7. PLOS authors have the option to publish the peer review history of their article (what does this mean?). If published, this will include your full peer review and any attached files.

Reviewer #2: No

---

## [Author Response · Author response to Decision Letter 2]

9 Jan 2020

Prof. Giuseppe Biagini,

Chief of Editor

PLOS ONE

January 9th, 2020

Dear Prof. Giuseppe Biagini,

Thank you very much for your e-mail of January/09/2020 with regard to our manuscript (PONE-D-19-25631R2) entitled “Epilepsy in a melanocyte-lineage mTOR hyperactivation mouse model: a novel epilepsy model” together with the comments from the editor and reviewer. We appreciate the editors and reviewers very much for their positive and constructive comments and suggestions on our manuscript. We have revised our manuscript accordingly. The alterations are referred in this response letter.

We hope that the revised manuscript meets with your approval. 

Sincerely,

Mari Wataya-Kaneda, M.D., Ph.D.

Department of Dermatology, Course of Integrated Medicine, Graduate School of Medicine, Osaka University, 2-2 Yamadaoka, Suita, Osaka, 565-0872, Japan

E-mail: mkaneda@derma.med.osaka-u.ac.jp

Tel: +81 6 6879 3031. Fax: +81 6 6879 3039

 

COMMENTS FROM EDITOR: 

“Thank you for submitting your manuscript to PLOS ONE. After careful consideration, we feel that it has merit but does not fully meet PLOS ONE’s publication criteria as it currently stands. Specifically, I invite you to complete and correct description of statistical analysis.” 

RESPONSE: Thank you for your important suggestions. We have carefully checked our manuscript again and corrected descriptions of statistical analysis in our revised manuscript. 

COMMENT: 1. “results of statistical analysis are not detailed in results, and should be better illustrated in legends to figures”

RESPONSE: Thank you for your kind advice. As requested, we have added the detail in legends to figures as indicated below. 

(Manuscript with Track Changes, Figure Legend section, page 24-26, line 488-549, in red with underline).

“Figure 1. Deletion of Tsc2 resulted in epilepsy in Tsc2Mitf-M cKO mice without obvious histoarchitectural changes.

A. Images captured from video recordings, showing typical spontaneous epilepsy in a 6-week-old cKO mouse. B. EEG and EMG segments (300 s) showing normal electrography in a control (WT) mouse and typical electrographic epilepsy in a cKO mouse. C. Relative brain and body weight in cKO mice compared with control (WT) mice at 9 and 11 weeks of age. *p < 0.05 and **p < 0.01 versus WT mice, n = 5 in each group, unpaired t-test. D. Hematoxylin staining of murine brain tissue sections, Scale bars: 600 µm. Sizes of hippocampus, Cerebral cortex and Whole brain are shown in the right panel, **p < 0.01 versus WT mice, n = 5 in each group, unpaired t-test. Data in C and D are expressed as mean ± SD. 

Figure 2. Hyperactivation of mTOR induced neural excitation in Tsc2Mitf-M cKO mice. 

A. Histoimmunostaining of whole brain sagittal sections from control (WT) mice (left panels) and cKO mice (right panels) at 5 weeks of age. p-S6 (upper panels) and c-FOS (bottom panels). The black rectangle outlines the area of hippocampus, cerebral cortex, and thalamus, and the detail is shown in the corresponding bottom panels. The circle shows representative p-S6 cytoplasmic and c-FOS nuclear positive staining. Scale bars: large bars, 600 µm; smaller bars, 200 µm. B. p-S6 and c-FOS positive rates (p-S6 or c-Fos-positive neuron cells versus all neuron cells) in the hippocampus, cerebral cortex, and thalamus. Data in C and D are expressed as mean ± SD. *P<0.05 versus WT mice, n = 5 in each group, unpaired t-test.

Figure 3. Histopathological analyses of the hippocampal region in Tsc2Mitf CKO mice. 

A. Immunofluorescence staining showing excitatory (CaMKII-α) and inhibitory (Parvalbumin) neurons in the hippocampus. The insets show higher magnification of positive cells (arrowheads). B. Numbers of CaMKII-α-positive cells and Parvalbumin-positive cells were double-blind counted in 10 random fields per tissue section. Ratio of inhibitory to excitatory neurons were calculated (n = 5 mice). Data in B are expressed as mean ± SD. n.s. means no significance versus WT mice, unpaired t-test. C. Timm staining. The amount of mossy fiber sprouting is similar in cKO and control (WT) mice. Scale bars: A, 200 µm; C, 200 µm. n.s., not significant.

Figure 4. Hyperproliferation of mitochondria in the neurons of Tsc2Mitf CKO mice. 

A. Morphologic examination of p-S6 high-expressed hippocampal CA1 pyramidal cells by electron microscopy. Enlarged cell bodies and mitochondria were observed in the neurons of cKO mice. Bottom panels represent high-magnification images of the regions designated by squares. B. Quantification of mitochondria. The number of mitochondria increased more than 5-fold and the mitochondrial size increased more than 2-fold in neurons from cKO mice, compared with control (WT) mice. (n = 20 cells/mouse, 3 mice in each group). Data in B are expressed as mean ± SD, unpaired t-test versus WT mice. C. Immunofluorescence staining showed hyperactivation of mTOR (p-S6) with hyperproliferation of mitochondria (COXIV) in the hippocampus of cKO mice. Scale bars: A upper panel, 2 µm; A bottom panel, 500 nm; C, 200 µm. 

Figure 5. Neurons from Tsc2Mitf -M cKO mice were more excitable than neurons from control mice.

A. Neurons were isolated from 4-week-old mice and cultured for 1 week. Immunofluorescence staining indicates that more than 80% of the isolated cells were neurons (GFAP, astrocytes; MAP2, neurons). B. Calcium imaging of cultured neurons, with corresponding traces shown at the bottom. Neurons from cKO mice respond to particularly low (10 mM) K+ stimulation. Scale bars: A, 100 µm; B, 100 µm. WT, wild-type.

Figure 6. Rapamycin treatment reduced seizures and number of mitochondria.

A. Frequency and duration of seizures (sz) in cKO mice in the absence or presence of rapamycin, n = 5 in each group. Data in A are expressed as mean ± SD. Unpaired t-test versus vehicle-treated mice. B. Histoimmunostaining analyses of mTOR signaling (p-S6), mitochondria (COXIV), and neuronal excitation (c-FOS). Scale bars: 100 µm. WT, wild-type. C. The fluorescence intensity was quantified by ImageJ. n = 5 in each group. Data in C are expressed as mean ± SD. One-way ANOVA test, followed by Dunnett's post hoc test for multiple comparisons (WT mice versus cKO mice; cKO mice versus Rapamycin-treated cKO mice (cKO_Rapa); WT mice versus cKO_Rapa mice) was performed and adjusted P values were calculated. *p < 0.05, **p < 0.01.

Supplementary information

S1 video. Video recording of typical epilepsy.”

COMMENT: 2. “in methods, I read that you used "two-way analysis of variance", but the post hoc test is not indicated.”

RESPONSE: Thank you for your careful reading and the important comment. Regret for this mistake, in our experiment, in Figure 6C, the differences of WT mice versus cKO mice, cKO mice versus Rapamycin-treated cKO mice, and WT mice versus cKO_Rapa mice were calculated using One-way ANOVA test, followed by Dunnett's post hoc test. As suggested, the description have been corrected in our revised manuscript as indicated below. 

(Manuscript with Track Changes, Figure Legend section, page 26, line 537-547, in red with underline; Methods section, page 9, line 148-152, in red with underline).

“Figure 6. Rapamycin treatment reduced seizures and number of mitochondria.

Frequency and duration of seizures (sz) in cKO mice in the absence or presence of rapamycin, n = 5 in each group. Data in A are expressed as mean ± SD. Unpaired t-test versus vehicle-treated mice. B. Histoimmunostaining analyses of mTOR signaling (p-S6), mitochondria (COXIV), and neuronal excitation (c-FOS). Scale bars: 100 µm. WT, wild-type. C. The fluorescence intensity was quantified by ImageJ. n = 5 in each group. Data in C are expressed as mean ± SD. One-way ANOVA test, followed by Dunnett's post hoc test for multiple comparisons (WT mice versus cKO mice; cKO mice versus Rapamycin-treated cKO mice (cKO_Rapa); WT mice versus cKO_Rapa mice) was performed and adjusted P values were calculated. *p < 0.05, **p < 0.01.”

(Manuscript with Track Changes, Figure Legend section, page 26, line 537-547, in red with underline; Methods section, page 9, line 148-152, in red with underline).

“Statistical analyses

Data are presented as mean ± SD. Unpaired Student’s t-test (Microsoft Excel; Microsoft Corp., Redmond, WA) was used for comparisons between two groups. One-way ANOVA test, followed by Dunnett's post hoc test was used for multiple comparisons (Microsoft Excel). P-values <0.05 were considered statistically significant.”

---

## [Editor Report · Decision Letter 3]

10 Jan 2020

Epilepsy in a melanocyte-lineage mTOR hyperactivation mouse model: a novel epilepsy model

PONE-D-19-25631R3

Dear Dr. Wataya-Kaneda,

We are pleased to inform you that your manuscript has been judged scientifically suitable for publication and will be formally accepted for publication once it complies with all outstanding technical requirements.

With kind regards,

Giuseppe Biagini, MD

Academic Editor

PLOS ONE
---

## [Editor Report · Acceptance letter]

16 Jan 2020

PONE-D-19-25631R3 

Epilepsy in a melanocyte-lineage mTOR hyperactivation mouse model: a novel epilepsy model 

Dear Dr. Wataya-Kaneda:

I am pleased to inform you that your manuscript has been deemed suitable for publication in PLOS ONE. Congratulations! Your manuscript is now with our production department. 

With kind regards,

on behalf of

Dr. Giuseppe Biagini 

Academic Editor

PLOS ONE